# WildSmoke: Ready-to-Use Dynamic 3D Smoke Assets from a Single Video in the Wild

Yuqiu Liu     Jialin Song     Manolis Savva     Wuyang Chen
Simon Fraser University
{yuqiu_liu, jialin_song, msavva, wuyang_chen}@sfu.ca

## Abstract

*We propose a pipeline to extract and reconstruct **dynamic 3D smoke assets** from a **single in-the-wild video**, and further integrate interactive simulation for smoke design and editing. Recent developments in 3D vision have significantly improved reconstructing and rendering fluid dynamics, supporting realistic and temporally consistent view synthesis. However, current fluid reconstructions rely heavily on carefully controlled clean lab environments, whereas real-world videos captured in the wild are largely underexplored. We pinpoint three key challenges of reconstructing smoke in real-world videos and design targeted techniques, including smoke extraction with background removal, initialization of smoke particles and camera poses, and inferring multi-view videos. Our method not only outperforms previous reconstruction and generation methods with high-quality smoke reconstructions (+2.22 average PSNR on wild videos), but also enables diverse and realistic editing of fluid dynamics by simulating our smoke assets. Project page:*
[*https://autumnyq.github.io/WildSmoke/*](https://autumnyq.github.io/WildSmoke/).

## 1. Introduction

Fluid phenomena, from eddies around high-speed trains to smoke rings from jet engines, are ubiquitous. A key challenge is reconstructing unobserved quantities such as velocity and density over the full spatiotemporal domain (3+1D) from visual inputs (2D photos or video). This task, formally inferring **3D fluid fields** from imagery, i.e., reconstructing fluid dynamics from visual observations, underpins applications from high-fidelity smoke rendering in visual effects [22, 41] to diagnostics in industrial flows [3, 31, 37]. Recent 3D vision advances have accelerated progress: multi-view benchmarks [10] provide accurately calibrated flow videos, and new methods infer fluid fields by jointly optimizing differentiable physics (via physics-based constraints) and neural scene representations (via rendering losses) from video [6, 8, 9, 13, 14, 45].

Despite notable advances, most existing approaches still rely heavily on **controlled multi-view** recordings for reconstructing fluid phenomena, which **differ substantially from real-world smoke in the wild**. Datasets such as ScalarFlow [10], TomoFluid [48], and FluidNexus [13] rely on fixed, precisely calibrated cameras, controlled smoke generation in laboratory settings, and pre-captured simplistic backgrounds. Such conditions are impractical for in-the-wild smoke, which is often recorded with a single moving camera (e.g., handheld or drone footage), with cluttered backgrounds and unpredictable camera motion.

At the same time, demand for ready-to-use 3D assets has surged across graphics and simulation applications [30, 42, 51].**Dynamic 3D smoke assets are particularly valuable** for downstream visual effects editing and real-time simulations [2, 5, 52], owing to their intricate motion and evolving volumetric properties. Yet, despite their importance, the generation of dynamic 3D smoke assets remains largely underexplored. We are motivated to ask:

> *Given a single in-the-wild video, how can we recover the underlying fluid field and enable downstream applications using the reconstructed smoke assets?*

In this work, we propose the **first complete, practically working pipeline** for reconstructing ready-to-use dynamic 3D smoke assets from **a single in-the-wild video**, a problem existing methods *have not addressed* and *fundamentally fail to solve*, as we empirically demonstrate. We are also the **first to formally identify** three core challenges of in-the-wild smoke reconstruction (Fig. 1, Section 2): noisy backgrounds, unknown camera poses, and coupled camera viewpoints and timesteps. Addressing these requires highly non-trivial, tightly integrated techniques: 1) in-context smoke segmentation combined with physically-grounded dehazing for light smoke, which no off-the-shelf tool handles jointly; 2) smoke-mask-filtered pose and particle initialization, without which training fails to converge; 3) spatiotemporal decoupling via multi-view generation and local pose perturbation. Furthermore, we are the *first to collect real-world wild smoke videos* and provide rigorous preliminary reconstruction studies on

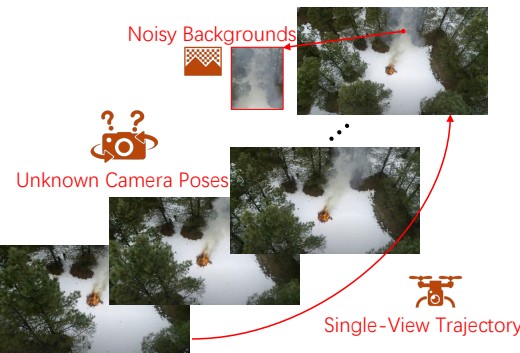

Noisy Backgrounds

Unknown Camera Poses

Single-View Trajectory

Figure 1. Challenges of smoke reconstruction from a single in-the-wild video: 1) noisy backgrounds and boundaries; 2) unknown camera poses; 3) single video with coupled camera viewpoints and timesteps.

them. We additionally demonstrate practical applications through fluid simulation. Our main contributions are summarized as follows:

1. We design a comprehensive pipeline that makes smoke reconstruction practical for unconstrained in-the-wild videos, extracting and reconstructing dynamic 3D smoke assets with consistent improvements over prior work (+2.22 dB PSNR on average on in-the-wild videos) (Section 4.3).

2. We comprehensively study alternative options for each component in our framework, and quantify the impact of each alternative on the final reconstruction performance (Section 3 and App. 8).

3. Our smoke assets are ready-to-use: we support realistic editing of smoke through interactive fluid simulations (Section 3.5 and 4.4).

## 2. Background

**Smoke Reconstruction from Videos.** Given a single-view video of upward-rising smoke, our goal is to reconstruct a coherent, view-consistent, and editable dynamic 3D smoke field over time $t = 1, \cdots, T$. Following FluidNexus [13], we represent smoke with physical particles $\{\mathbf{p}_t^{\text{phy}}, \mathbf{u}_t\}$ for density/velocity simulation and visual Gaussian particles $\{\mathbf{p}_t^{\text{vis}}, \mathbf{c}_t, \mathbf{s}_t, \mathbf{o}_t, \mathbf{r}_t\}$ for differentiable rendering. The density field $\rho_t$ and velocity field $\mathbf{V}_t$ can be mapped from particles to grids via kernel-weighted interpolation [13, 18–20, 29].

**Importance and Challenges.** While smoke can be simulated with strong artistic control, simulation alone is underconstrained to match a specific real shot and its scene interactions. Reconstructing smoke from in-the-wild videos instead recovers scene-conditioned density and velocity fields that can initialize and steer downstream simulation-based editing. However, this setting introduces three challenges absent from controlled smoke captures. First, semi-transparent smoke reveals **complex backgrounds and noisy bound-**

**aries**, unlike ScalarFlow [10], TomoFluid [48], and FluidNexus [13], which rely on controlled or post-processed backgrounds. Second, wild videos are often captured by moving handheld or drone cameras with **unknown poses**. Third, a single video provides only one camera trajectory over time, creating an approximate one-to-one **coupling between viewpoints and timesteps**. Our pipeline explicitly addresses these challenges and outperforms existing smoke reconstruction methods [13, 45].

## 3. Methods

To address the above challenges, we propose a unified pipeline to reconstruct clean, background-free smoke from unconstrained real-world videos, as shown in Figure 2.

### 3.1. Smoke Extraction

In real-world scenarios, smoke often exhibits complex, irregular boundaries and is intertwined with significant noise from backgrounds. To address this issue, we propose a pipeline that extracts clean smoke regions from a single in-the-wild video.

**(a) Smoke Mask Extraction.** We first segment the smoke and obtain a binary mask sequence $\mathbf{M} = \{M_t\}$ over all timesteps $t$ in the video. Owing to the *widely observed limitation* of off-the-shelf semantic segmentation models, which generally struggle to generalize to in-the-wild smoke, we adopt a one-shot learning strategy, using SAM [23] for annotation and SegGPT [40] for propagation[1]. For each video, we select an early reference frame $I_{\text{ref}}$, interactively annotate its smoke with SAM to obtain $M_{\text{ref}}$[2], and then feed the pair $(I_{\text{ref}}, M_{\text{ref}})$ to SegGPT [40] to perform one-shot inference and produce masks $M_t$ for the remaining frames $I_t$. This combines SAM's high-quality per-frame annotation with SegGPT's frame-wise generalization.

The boundary of segmented smoke could still be noisy. We apply a Gaussian filter $f_g$ to smooth the boundary of inferred masks, yielding a more natural smoke contour. The smoke areas are extracted via $\tilde{S}_t = f_g(M_t) \cdot I_t$. As shown in Figure 3(a), this in-context segmentation can reliably segment smoke boundaries. See Supplement 9.3 in the supplement for more smoke segmentation examples.

At this stage, it is necessary to further distinguish dense from light smoke. We categorize them based on the visibility of the background. For dense smoke, the background is completely occluded due to high optical density (low transmittance). In this case, background contamination is negligible, and the smoke can be directly treated as the masked region. However, for the light smoke, the background remains partially visible (nonzero transmittance). Here, a dehazing step

---

[1]Although it is possible to fine-tune a segmentation model for general smoke extraction, the required data collection and annotation effort is beyond the scope of this work.

[2]We use an online SAM-based annotator https://roboflow.com/.

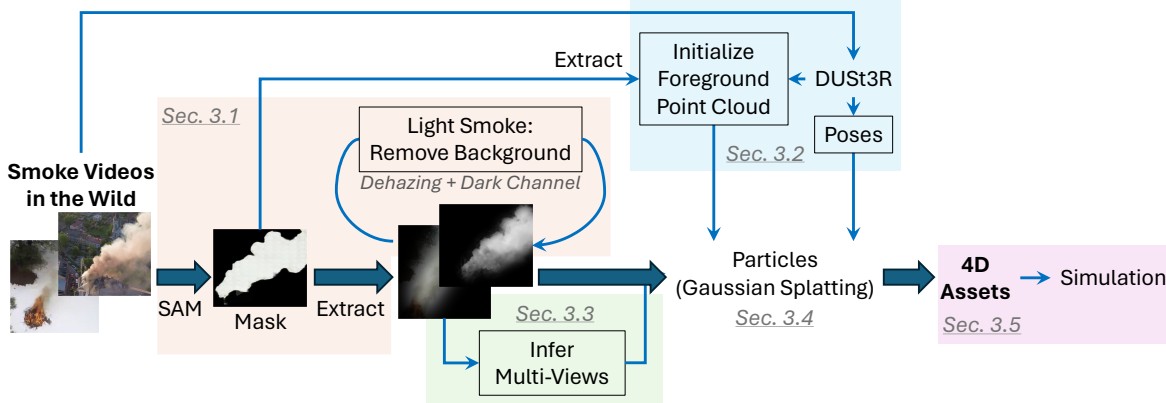

Figure 2. To reconstruct smoke from a single video in the wild, our pipeline includes five steps: 1) Smoke extraction, with background removed for light smoke due to translucence (Section 3.1); 2) Pose estimation and coarse initialization for 3D point cloud (Section 3.2); 3) Inferring multi-views for smoke (Section 3.3); 4) Training smoke particles (Section 3.4); 5) composition and simulation of 4D smoke assets (Section 3.5). "SAM": segment anything [23]. "DUSt3R" [39].

is required to remove background leakage before smoke reconstruction.

**(b) Dehazing for Background Removal.** Similar to fog or haze, background objects remain visible through light smoke due to light transmission, introducing artifacts during reconstruction. For light smoke, we fine-tune a pretrained DehazeFormer [35] to isolate the smoke and remove the background. We construct a fine-tuning dataset of synthetic smoky inputs $\tilde{I}$ (pixels normalized to $[0, 1]$) by blending coarsely extracted smoke $\tilde{S}$ with video frames of clean background without smoke $I^{\text{clean}}$, which serve as supervision targets:

$$\tilde{I} = I^{\text{clean}} \cdot \tilde{T} + A \cdot \tilde{S}, \quad (1)$$

where $\tilde{T} = 1 - \tilde{S}$ is the coarse transmission map, representing the amount of background light that reaches the camera through the smoke. We estimate the atmospheric light $A$ using the dark channel prior [15]. As shown in Figure 3 (c), the fine-tuned DehazeFormer can remove the foreground smoke and recover the background $\tilde{I}^{\text{clean}}$.

For dense smoke where background contamination is negligible, the extracted smoke $\tilde{S}$ from masks can be used directly without this dehazing step.

**(c) Extract Clean Smoke.** For light smoke, we extract the clean foreground smoke based on Equation (1):

$$S = 1 - \frac{I - A}{\tilde{I}^{\text{clean}} - A}, \quad (2)$$

where $I$ is the smoky frame in the original video, and $\tilde{I}^{\text{clean}}$ is the background recovered by DehazeFormer. The resulting foreground-smoke images are then used for smoke-field reconstruction.

**Alternative Approach: Mask-to-Matte Refinement.** We also provide an alternative for dehazing. We refine the coarse binary masks into pixel-accurate alpha mattes using Video-MaMa [25], a mask-guided video matting model that converts masks to alpha.

We apply VideoMaMa to the coarse smoke mask sequence $\{M_t\}$ and obtain an alpha matte sequence $\{\alpha_t\}$, where $\alpha_t \in [0, 1]$ encodes smoke opacity. We then use $S = \alpha_t * I_t$ as our grayscale smoke observation for all subsequent steps (i.e., background-free smoke input to reconstruction).

## 3.2. Pose Estimation and Coarse Geometry

To obtain camera intrinsics and extrinsics for frames, we initialize camera poses and focal lengths with estimates from the pretrained DUSt3R [39]. We further initialize both physical and visual particles from a sparse 3D point cloud $\{\mathbf{p}_i\}_{i=1}^N$ estimated by DUSt3R, which provides a rough geometry of the scene, including both smoke and background regions. We filter the DUSt3R-generated 3D point cloud using the extracted smoke masks, retaining only the smoke-relevant points to initialize our particles. Although DUSt3R produces coarse results, this initialization provides meaningful spatial priors for optimizing our particles; without it, training may fail to converge. See more details in Supplement7.2 in the supplement.

**Alternative Approach: VGGT [38].** This initialization can be replaced by any pose estimation work pretrained on outdoor datasets. We also test the results on VGGT [38] (See App. 8 for experiments and more details).

## 3.3. Inferring Multi-View Videos

In wild smoke videos, cameras typically follow a single trajectory. For example, a drone may fly from the ground upwards along with the smoke plume to capture the video. Although the camera may span a wide spatiotemporal extent, camera viewpoints and timesteps are highly coupled,

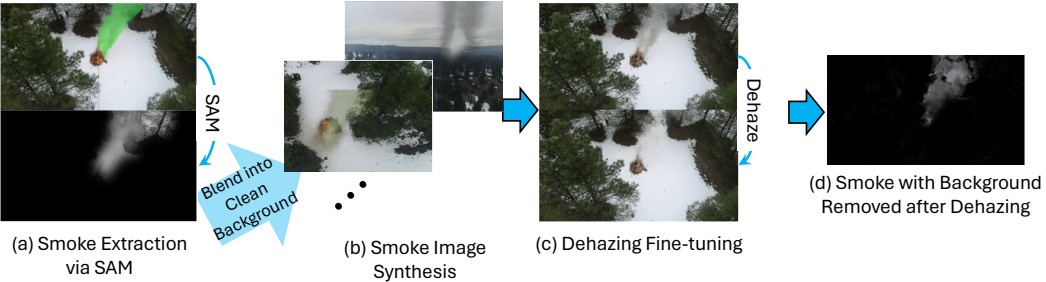

(a) Smoke Extraction via SAM

SAM

Blend into Clean Background

(b) Smoke Image Synthesis

(c) Dehazing Fine-tuning

Dehaze

(d) Smoke with Background Removed after Dehazing

Figure 3. Smoke extraction, with background removal for light smoke.

i.e., each camera viewpoint is associated with a unique timestep, and vice versa. This coupling can cause overfitting and degrade novel-view synthesis for unseen viewpoints or timesteps.

To decouple the spatiotemporal camera trajectory, in our work we employ generative multi-view synthesis. To obtain multi-view supervision from single-view input, we use pre-trained SV4D 2.0 [44] to generate smoke videos from novel views. We choose azimuth angles of $[-10°, 10°, 20°, 30°]$ relative to the pose of the current frame as our novel viewpoints. Since SV4D 2.0 does not support long temporally consistent video generation, we split our video sequences into short clips, and overlap one frame between neighboring clips, ensuring valid conditioning for every novel-view segment. Per-frame camera-to-world poses are obtained by applying the angle set to the DUSt3R-initialized poses.

Moreover, to address the unreliable frames produced by SV4D at later timesteps, we apply an exponentially decaying weight over the frame index. This strategy progressively down-weights the influence of later frames generated by SV4D 2.0 during Gaussian-particle training. See Supplement 7.4 and 7.5 for more details.

**Alternative Approach:** The multi-view generator can be replaced by other novel-view video synthesis models, e.g., TrajectoryCrafter [46]. However, for outdoor videos with cluttered backgrounds, their quality degrades severely under large viewpoint changes and long sequences. We find SV4D 2.0 remains the most reliable drop-in choice after background removal. See examples in App. 8.

### 3.4. Training Gaussian Particles

Following the training strategy of FluidNexus [13], we separately train visual and physical particle representations. Particles are optimized by minimizing both visual supervision (photometric errors between input frames and rendered views using 3D Gaussian Splatting [21]) and also physical regularization based on fluid simulation. To overcome the *"numerical diffusion"* during fluid simulation that leads to artificial smoothing [1], we introduce a frequency-aware regularization during training. Moreover, to determine the *unknown buoyancy in the wild*, we make its strength learnable. For more training details, we refer readers to Supplement 7.6.

To further disentangle spatial and temporal viewpoints in single-camera trajectories from in-the-wild videos, we not only train with our generated multi-view trajectories (Section 3.3), but also progressively enrich the view trajectory by **perturbing local camera poses**. Specifically, after the particles at time $t$ converge, we introduce perturbed viewpoints by shifting the camera pose forward by $\Delta t$ along the trajectory (modulo the sequence length $T$). Concretely, together with the original pose $(\mathbf{R}_t, t)$, we also include $(\mathbf{R}_{(t+\Delta t) \bmod T}, t)$ as an input and its corresponding rendering result as the target. Here, $\mathbf{R}$ is the rotation matrix of the camera's extrinsics; $\Delta t \ll T$, i.e., we perturb within a very short period relative to the whole temporal domain. Essentially, we effectively decouple viewpoint and timestep by associating pose $\mathbf{R}_{(t+\Delta t) \bmod T}$ with timestep $t$ via "local pose perturbation." Since $\Delta t$ is small, perturbed novel viewpoints still reside in neighborhoods of the original video trajectory. In practice, we progressively increase $\Delta t$ from 2 to 4 during training.

**Alternative Approach.** We also consider HyFluid [45] and FluidNexus [13], but both are less suitable for in-the-wild videos in our setting. FluidNexus assumes a fixed buoyancy term and lacks our spatiotemporal augmentation (local pose perturbation), which can lead to incorrect force balance and poorer reconstruction, which motivates our learnable buoyancy and augmentation design. Detailed comparisons are reported in App. 8.

### 3.5. Ready-to-Use 4D Smoke Assets

A key use case of our reconstructed smoke assets is visual-effects (VFX) editing of novel smoke scenes. In our work, we consider simulating smoke interactions using Phi-Flow [16]. The simulation is initialized with the reconstructed density field $\rho$ from visual particles, and velocity field $\mathbf{V}$ from physical particles:

**Density Field from Visual Particles.** For each visual particle $i$ with center $\mathbf{p}_{t,i}^{\text{vis}}$, scale $\mathbf{s}_{t,i}$, rotation $\mathbf{r}_{t,i}$, and opacity $\mathbf{o}_{t,i}$, we define a Gaussian kernel

$$\phi_{t,i}(\mathbf{x}) = \exp\left(-\frac{1}{2}(\mathbf{x} - \mathbf{p}_{t,i}^{\text{vis}})^\top \mathbf{\Sigma}_{t,i}^{-1} (\mathbf{x} - \mathbf{p}_{t,i}^{\text{vis}})\right), \quad (3)$$

where $\mathbf{\Sigma}_{t,i} = R(\mathbf{r}_{t,i}) \operatorname{diag}(\mathbf{s}_{t,i}^2) R(\mathbf{r}_{t,i})^\top$, $\mathbf{r}_{t,i} \in \mathbb{R}^4$ is the unit quaternion representing the particle rotation, and

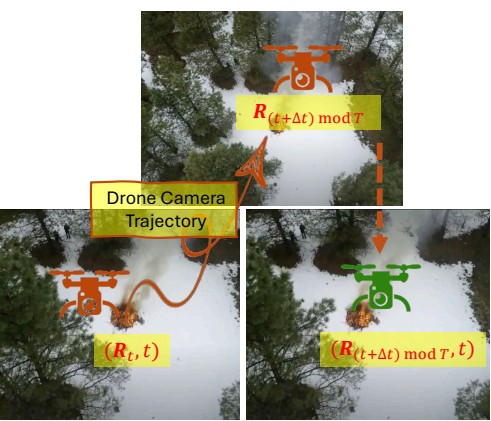

Figure 4. Local pose perturbation. Given the original camera pose $(\mathbf{R}_t, t)$ on a single-video trajectory ("red drone"), we additionally introduce a perturbed viewpoint by shifting the camera pose by $\Delta t$ along the trajectory with wrap-around $(t + \Delta t)$ mod $T$ ("green drone"). For this perturbed viewpoint, the rendered image serves as supervision. Both original and perturbed samples are used during following training.

$R(\mathbf{r}_{t,i}) \in \mathbb{R}^{3 \times 3}$ is the corresponding rotation matrix obtained from $\mathbf{r}_{t,i}$. The density field is the weighted sum of all particle kernels:

$$\rho_t(\mathbf{x}) = \sum_{i=1}^{N_t^{\text{vis}}} \mathbf{o}_{t,i} \, \phi_{t,i}(\mathbf{x}). \qquad (4)$$

**Velocity Field from Physical Particles.** Given a physical particle with position $\mathbf{p}_{t,i}^{\text{phy}}$ and velocity $\mathbf{u}_{t,i}$, we map them to the grid. The Gaussian kernel $\phi'_{t,i}$ for the velocity field has the same form as $\phi_{t,i}$ but is centered at $\mathbf{p}_{t,i}^{\text{phy}}$:

$$\mathbf{V}_t(\mathbf{x}) = \frac{\sum_{i=1}^{N_t^{\text{phy}}} \phi'_{t,i}(\mathbf{x}) \, \mathbf{u}_{t,i}}{\sum_{i=1}^{N_t^{\text{phy}}} \phi'_{t,i}(\mathbf{x}) + \varepsilon}, \qquad (5)$$

here we use $\varepsilon = 10^{-8}$. To ensure spatial consistency between the density and velocity fields, both visual and physical splatting are restricted to the same bounding region defined by the visual particles.

After the initialization of density and velocity fields, we continue the fluid dynamics in PhiFlow [16] with the standard MacCormack semi-Lagrangian method [32] for advection and the projection-based pressure-Poisson solver for incompressible flow. We consider either external wind forces or an inserted obstacle to interact with the smoke.

## 4. Experiments

### 4.1. Settings

**Datasets.** We evaluate our pipeline on both synthetic and real-world collections of smoke, assessing the reconstruction quality using view rendering. Note that, in the **absence** of any existing large-scale benchmark dedicated to real-world smoke videos, we undertake a *substantial data collection effort* and construct our dataset from scratch.

- We **synthesize** two high-resolution and photorealistic smoke videos by combining a 4D smoke VDB sequence with a real 3D scene. These paired videos, rendered with distinct camera trajectories, allow direct comparison between our reconstructed results and the ground-truth. Specifically, we download a synthetic 4D smoke VDB sequence and a 3D scene from CGTrader[3], combine them and render two videos in Blender[4] with two camera trajectories. One trajectory is used for training and another for evaluation. See Supplement 7.1 for more rendered samples.
- We also evaluate on two **real-world** testbeds: 1) FLAME dataset [33] is a fire-imaging dataset collected by drones during a prescribed burning of piled detritus in an Arizona pine forest. The dataset includes video recordings and thermal heatmaps captured by infrared cameras. 2) We further collect three videos of smoke from Pixabay[5], covering diverse scenarios.

To unify our training and evaluation settings, we standardize all our videos to 270 frames; training timesteps are $t = 1, \ldots, T$ with $T = 240$, and unseen future timesteps are $t = T + 1, \ldots, T'$ with $T' = 270$.

We measure view changes along the video trajectory via the camera's relative rotation angle $\Delta\theta_{t_1, t_2} = \arccos\left(\frac{\text{tr}(\Delta\mathbf{R}_{t_1,t_2}) - 1}{2}\right)$, where $\Delta\mathbf{R}_{t_1,t_2} = \mathbf{R}_{t_2}^\top \mathbf{R}_{t_1}$ ($\mathbf{R}$ is the rotation matrix of the camera's extrinsics). In our synthetic data setting, the camera undergoes a rotation of $53°$ from $t = 1$ to $t = T$, which is larger than the $7°$ rotation during the future steps from $t = T + 1$ to $t = T'$. Smoke in our videos is always centered in the scene.

**Tasks.** As explained by $\Delta\theta_{1,T}$ and $\Delta\theta_{T+1,T'}$ above, larger pose changes occur within the training timesteps than in the future steps. Synthesizing frames from the fixed viewpoint at the beginning of video ($t = 1$) is more challenging than following the camera pose trajectory.

Therefore, in our evaluation, the **novel view** is defined with camera pose at $t = 1$, and the **input view** follows the ground-truth camera trajectory.

Following [45], we consider the following two tasks:

- **Novel view synthesis:** We render smoke views from the fixed novel viewpoint over the training timesteps. Specifically, we fix the camera pose at $t = 1$ and synthesize novel views through $t = 2, \cdots, T$. We study novel view

---

[3] https://www.cgtrader.com/3d-models
[4] https://www.blender.org
[5] Pixabay shares royalty-free images, videos, audio, and other media. All content is released by Pixabay under the Content License, which makes it safe to use without asking for permission or giving credit to the artist. See their license at https://pixabay.com/service/terms

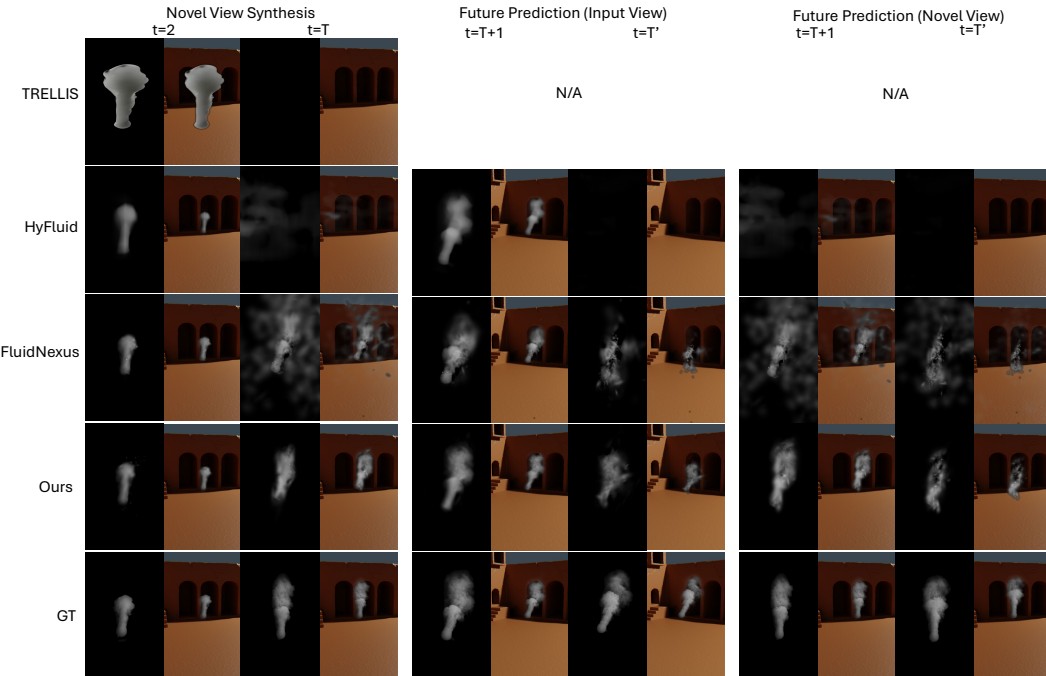

Figure 5. Visualization of novel view synthesis and future predictions on synthetic smoke videos. *Novel view* uses the camera pose at $t = 1$, and *input view* means the camera poses along the source video. Training uses $T = 240$ frames, and the unseen future extends to $T' = 270$ frames. *GT*: Ground Truth.

synthesis only on our synthetic dataset, due to the lack of ground-truth multi-view videos on real-world videos (FLAME and Pixabay).

- **Future prediction:** We extrapolate the fluid dynamics into the future steps $t = T + 1, \cdots, T'$. No model is ever trained with ground-truth future frames from videos. On synthetic videos, we study both quantitative and visual results, and predict futures for both input view (i.e. follow the ground-truth camera pose trajectory during $t = T + 1, \cdots, T'$) and novel view (fixed camera pose at $t = 1$). Due to the lack of ground-truth multi-view videos, on real-world videos (FLAME and Pixabay), we can only study future predictions based on the input view.

During inference, the learned velocity field is used to advect (evolve) the visual particles for future prediction. We refer the reader to [45] for more details about these tasks.

**Evaluation Metrics.** We report the peak signal-to-noise ratio (PSNR) averaged over frames, and defer the *structural similarity index measure (SSIM)* and the *perceptual metric LPIPS* [49] in Appendix 9.5. These metrics are *widely adopted* in prior smoke reconstruction [13, 45] and deblurring works [11, 12].

While perfectly faithful recovery of turbulent smoke from a single in-the-wild video is arguably ill-posed, and novel-view PSNR from a single input trajectory cannot fully resolve the resulting ambiguities, we design our method and evaluation protocol to provide, to the best of our knowledge, the most faithful reconstructions currently attainable under these challenging conditions.

**Baselines.** We compare with both reconstruction and generation methods, including HyFluid [45], FluidNexus [13], and Trellis [42]. We follow FluidNexus to train with the default grayscale input setting.

Table 1. Comparing PSNR (higher is better) of smoke reconstruction by different methods on the synthetic dataset. *Novel view* uses the camera pose at $t = 1$, and *input view* means the camera poses along the source video.

| Methods | Novel View Synthesis | Future Prediction (Input View) | Future Prediction (Novel View) |
|---|---|---|---|
| Trellis [42] | 19.98 | - | - |
| HyFluid [45] | 24.26 | 22.54 | 22.18 |
| FluidNexus [13] | 29.26 | 23.61 | 21.54 |
| Ours | **29.78** | **25.26** | **25.04** |

Table 2. Cumulative ablation study (PSNR) on the synthetic smoke video. *Novel view* uses the camera pose at $t = 1$, and *input view* means the camera poses along the source video.

| | Novel View Synthesis | Future Prediction (Input View) | Future Prediction (Novel View) |
|---|---|---|---|
| Baseline | 22.55 | 16.69 | 18.17 |
| + Smoke Extraction | 29.26 | 23.61 | 21.54 |
| + DUSt3R Init. | 29.48 | 26.45 | 22.92 |
| + Local Perturbation | 29.77 | **26.85** | 23.59 |
| + Multi-Views (Ours) | **29.78** | 25.26 | **25.04** |

## 4.2. Synthetic Data

**Results.** We first show quantitative results in Table 1 and visualizations in Figure 5. Trellis [42] fits a static 3D asset per frame (image-to-3D), estimating neither time-varying particles nor velocities; hence future prediction is ill-defined. Moreover, per-frame inconsistency in the novel-view synthesis setting causes the smoke to progressively fade, yielding a black frame at time $T$. Compared with HyFluid [45] and FluidNexus [13], our pipeline achieves higher PSNR and more stable visualizations.

**Ablation Study.** We further provide ablation studies in Table 2; see visualization results in Supplement9.7. Smoke extraction (segmentation), initializing poses/particles, and adding inferred multi-view supervision progressively improve PSNR and visual quality. Note that, due to significant spatiotemporal deviations, the task of *future prediction at novel view* is substantially more challenging than either *novel view synthesis* or *future prediction at input view*. As a result, *future prediction at novel view* demands additional geometric cues, even coarse ones generated by SV4D, and therefore benefits most (+1.45 PSNR from 23.59 to 25.04) from incorporating "+ Multi-Views."

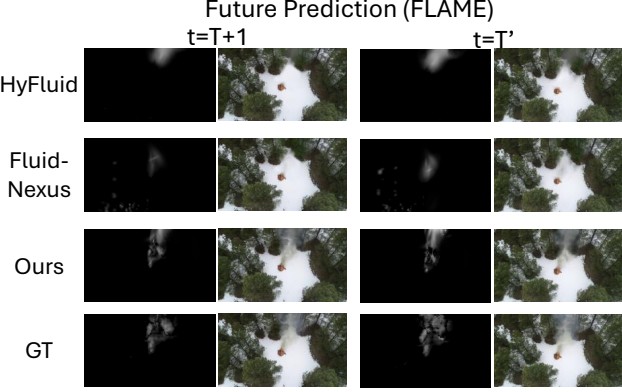

Figure 6. Visualization of future predictions (input view) on the FLAME dataset [33]. Training uses $T = 240$ frames, and the unseen future extends to $T' = 270$ frames. *GT*: Ground Truth.

Table 3. Comparing PSNR (higher is better) of smoke reconstruction by different methods on the FLAME dataset [33].

|  | Future Prediction (Input View) |
| --- | --- |
| HyFluid [45] | 21.67 |
| FluidNexus [13] | 21.78 |
| Ours | **22.88** |

## 4.3. Smoke Videos in the Wild

We further evaluate on in-the-wild smoke videos. Due to the lack of ground-truth novel views in real videos, here we evaluate only the task of *future prediction at input view*, meaning the camera poses along the source video.

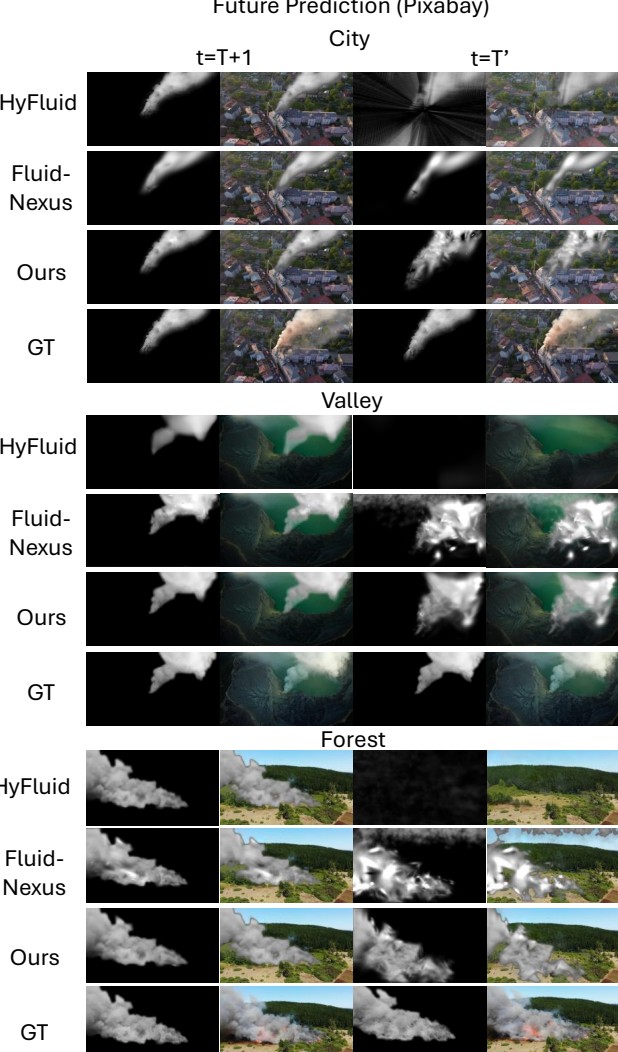

Figure 7. Visualization of future predictions (input view) on videos collected from Pixabay. Training uses $T = 240$ frames, and the unseen future extends to $T' = 270$ frames. *GT*: Ground Truth.

Table 4. Comparing PSNR (higher is better) of smoke reconstruction by different methods on videos collected from Pixabay.

|  | Future Prediction (Input View) | | |
| --- | --- | --- | --- |
|  | City | Valley | Forest |
| HyFluid [45] | 23.24 | 12.43 | 13.70 |
| FluidNexus [13] | 24.18 | 16.44 | 14.63 |
| Ours | **24.68** | **20.42** | **17.91** |

**FLAME [33].** The FLAME dataset includes drone-captured videos in the wild forest. The smoke is light, and thus background removal is necessary. We show the results in Table 3 and Figure 6.

**Pixabay.** We evaluate three thick-smoke videos from Pixabay. We show the results in Table 4 and Figure 7.

Overall, across both light and dense smoke over diverse real-world videos, our method outperforms prior work, achieving an average PSNR improvement of +2.22 dB. Qualitatively, the reconstructions remain consistent and blend back into the original backgrounds.

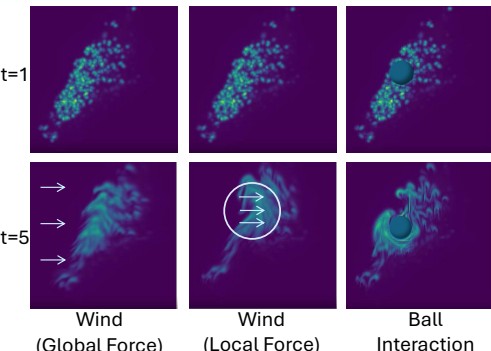

Figure 8. Visualizations of simulations with our smoke assets in different interaction scenarios. $t$ means simulation time, different from the video frames we used in previous figures. The simulation starts with particles reconstructed at the last frame of the "city" video from Pixabay (Figure 7 top).

### 4.4. Interactive Simulations

Finally, we demonstrate interactive simulations enabled by our fluid reconstruction. We consider two scenarios: (i) external wind (both global and local forces), and (ii) a rigid spherical obstacle. For the global wind, we apply a global constant force $f_w = (0.005, 0, 0)$ uniformly across the domain. The local wind uses the same magnitude and direction but is confined to a sphere of radius 30 at the scene center. For the obstacle case, we place a rigid ball (radius 10) centered at $(50, 70)$ and simulate the resulting interactions. All simulations use the reconstruction from the "city" video from Pixabay (Figure 7 top); results are shown in Figure 8 (XY-plane from Z-axis). We can see that our smoke assets support realistic editing via diverse simulation scenarios.

### 4.5. Training and Inference Cost

Table 6 in the Supplement 7.7 reports the per-stage GPU hours. Across datasets and resolutions, our pipeline reconstructs smoke from in-the-wild videos in 4.47 GPU-hours on average among all 5 videos. Notably, with the DUSt3R-based pose initialization, smoke can be localized without seeding a large number of particles, reducing the Gaussian-particle training stage by 1.5 GPU-hours.

### 5. Related Works

**Fluid Field Extraction from Videos.** Recent methods reconstruct fluid density and velocity from videos using neural representations and physical constraints. PINF [6], NeuroFluid [14], and HyFluid [45] use neural fields or particle-

based representations, while FluidNexus [13] incorporates generative priors for single-video smoke reconstruction. DiffWind [24] recovers wind force fields and object dynamics from video via differentiable rendering with Lattice Boltzmann Method constraints. However, they are mostly designed or evaluated in controlled settings with calibrated cameras, clean backgrounds, or lab-specific priors. We instead target single in-the-wild smoke videos and recover dynamic 3D smoke assets for simulation-based editing.

**Physics-based Fluid Simulation and Editing.** Eulerian [36] and Lagrangian [4] solvers enable fluid editing via forces, boundary conditions, and solid–fluid interactions. Differentiable systems such as DiffTaichi [19] and PhiFlow [16] further integrate these interactions into learning pipelines. PIDG [17] embeds constitutive laws into Gaussian Splatting by treating each particle as a Lagrangian material point supervised by optical flow. Unlike works assuming known simulated fields, we reconstruct density and velocity from in-the-wild videos and export simulation-ready smoke assets.

**In-the-Wild Physical Phenomena Synthesis.** RainyGS [7] and FieryGS [34] synthesize realistic rain and fire effects in open-world scenes by coupling 3D Gaussian Splatting with physics-based simulation. Similarly, we target in-the-wild settings but focus on recovering physically grounded smoke representations suitable for downstream simulation and editing.

**4D Generation.** Generative models have advanced image-to-3D, novel-view synthesis, and virtual-camera video generation [26, 47, 53]. SV4D [43] and SV4D 2.0 [44] improve spatiotemporal consistency for dynamic scenes, while GaussianPrediction [50] enables free-view future synthesis via motion extrapolation over 3D Gaussians. We use such generative consistency as auxiliary multi-view supervision, while reconstructing an explicit dynamic smoke representation with density, velocity, and simulation-based editability.

## 6. Conclusion

In this paper, we addressed the challenge of recovering time-varying 3D smoke fields from a single in-the-wild video and producing ready-to-use dynamic assets. The proposed pipeline tackles three core challenges—noisy backgrounds, unknown camera poses, and coupled spatiotemporal views—through one-shot segmentation and dehazing, pose estimation, and decoupled spatiotemporal trajectory through multi-view generation and local pose perturbation. We further showed that the reconstructed assets support physically consistent editing through fluid simulation, aligning visual fidelity with physical plausibility and supporting downstream simulation workflows.

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

# WildSmoke: Ready-to-Use Dynamic 3D Smoke Assets from a Single Video in the Wild

## Supplementary Material

## 7. Technical Details

### 7.1. Synthetic Data

To construct the synthetic dataset for our evaluation on novel view synthesis, we obtain a 4D smoke sequence and a static 3D scene from CGTrader. The sequence is placed into the 3D scene and rendered along a prescribed camera path. For the training video, the camera pitch is fixed at $\theta_p = 10°$, and the azimuth is swept linearly from $-105°$ to $-45°$ at a constant speed:

$$\phi(t) = -105° + 60° \cdot \frac{t}{T'}, \quad t \in [0, T'], \qquad (6)$$

where $T' = 270$. Rendering is performed with Blender *Cycles* (max samples = 200, i.e., each pixel is estimated by averaging up to 200 light paths to reduce Monte Carlo noise). Representative frames are shown in Figure 9. In the testing video for the novel view synthesis, the camera is fixed at $\phi(t = 0) = -105°$ and $\theta_p = 10°$, i.e., the camera pose at the first frame. During inference, we use the pose estimated by DUSt3R at the first frame from the training video as the input pose.

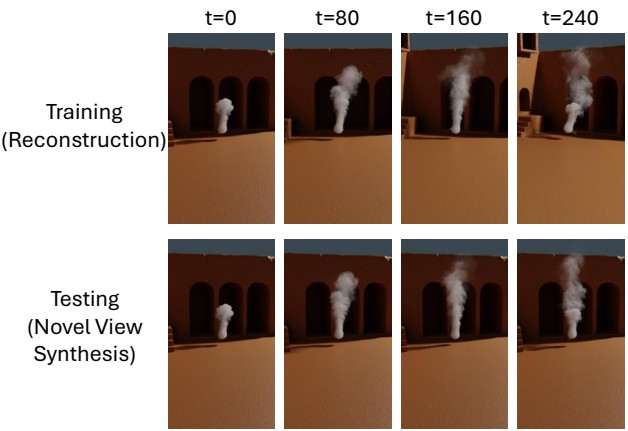

Figure 9. Frames in our synthetic smoke video. When training the reconstruction, the camera view is moving along a trajectory defined by Equation 6. During testing, we evaluate the novel view synthesis with the fixed camera pose from the first frame ($\phi(t = 0)$).

### 7.2. Pose Estimation

We infer per-frame camera poses with a pretrained DUSt3R in the one-reference mode, using the first frame as reference. Since DUSt3R and Gaussian Splatting (GS) use differ-

ent conventions (GS looks along $-Z$), we convert DUSt3R camera-to-world poses by *negating* the $y$ and $z$ axes (i.e., multiply both by $-1$). An example of estimated poses and point cloud is shown in Figure 10. Formally, for each DUSt3R-predicted pose $\mathbf{C}$, we apply

$$\mathbf{C}' = \mathbf{F}\,\mathbf{C}, \quad \mathbf{F} = \mathrm{diag}(1, -1, -1, 1),$$

where $\mathbf{F}$ flips the $y$ and $z$ axes.

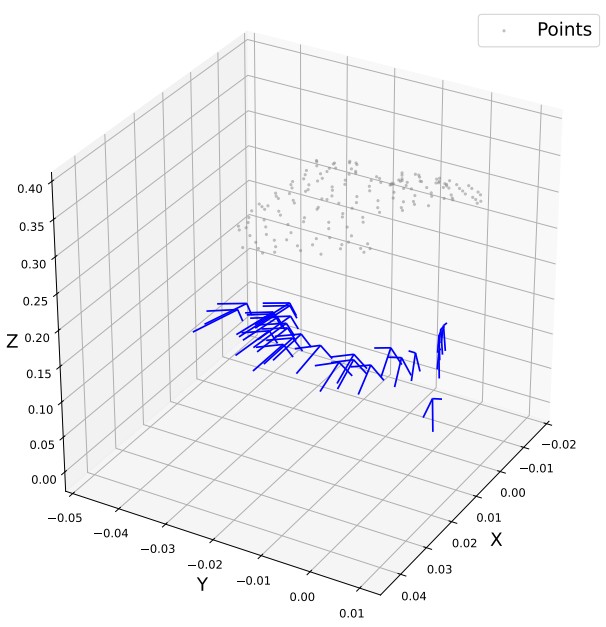

Figure 10. Camera poses and foreground smoke points (for the "valley" video in Figure 7 (middle) from Pixabay) predicted by DUSt3R. Blue arrows are DUSt3R camera-to-world poses, gray dots are the reconstructed foreground point clouds.

### 7.3. Particle Initialization

To accelerate convergence, we initialize both physical and visual particles from the foreground point cloud predicted by DUSt3R. Let $\mathbf{X} \in \mathbb{R}^{T \times H \times W \times 3}$ denote the 3D points (camera-to-world coordinates) for all frames, where $\mathbf{X}_{t,h,w} \in \mathbb{R}^3$ is the 3D position of pixel $(h, w)$ at time $t$. We obtain a reliable smoke foreground by intersecting the segmentation masks $M_t$ with the DUSt3R confidence mask $M_t^{\mathrm{conf}}$, and collect the foreground set:

$$\mathcal{P} = \left\{ \mathbf{x} \in \mathbb{R}^3 \,\middle|\, \mathbf{x} = \mathbf{X}_{t,h,w}, \ M_t(h, w) \wedge M_t^{\mathrm{conf}}(h, w) = 1 \right\}.$$

As DUSt3R and Gaussian Splatting (GS) follow different axis conventions (GS looks along $-Z$), we convert point coordinates by *negating* the $y$ and $z$ axes (multiply both by $-1$). With $\mathbf{F}' = \mathrm{diag}(1, -1, -1)$, we define

$$\tilde{\mathcal{P}} = \left\{ \mathbf{F}' \mathbf{x} \mid \forall \mathbf{x} \in \mathcal{P} \right\}.$$

After that, we downsample $\tilde{\mathcal{P}}$ with a voxel grid, which merges points falling into the same 3D cell into a single representative, to control the initial particle count. We retain 100–300 foreground points per video. The coordinates of the points are then used to initialize our physical and visual particles. After training, the physical and visual particle counts typically grow to approximately 5–10k.

## 7.4. Generated Videos

SV4D 2.0 takes yaw/pitch in radians, so we recover each frame's camera pose by applying the generation-time angle offsets to the DUSt3R pose, while using the DUSt3R-predicted point cloud center as the fixed look-at target. Let the DUSt3R pose of the input frame be $\mathbf{P}_0 = [\mathbf{R}_0 \mid \mathbf{t}_0] \in \mathbb{R}^{3 \times 4}$, and let $\mathbf{c} \in \mathbb{R}^3$ be the scene center estimated from the DUSt3R point cloud. Given per-frame angle offsets (in radians) $\Delta\phi_t$ (yaw/azimuth) and $\Delta\theta_t$ (pitch), define

$$\mathbf{R}_y(\Delta\phi) = \begin{bmatrix} \cos\Delta\phi & 0 & \sin\Delta\phi \\ 0 & 1 & 0 \\ -\sin\Delta\phi & 0 & \cos\Delta\phi \end{bmatrix},$$

$$\mathbf{R}_x(\Delta\theta) = \begin{bmatrix} 1 & 0 & 0 \\ 0 & \cos\Delta\theta & -\sin\Delta\theta \\ 0 & \sin\Delta\theta & \cos\Delta\theta \end{bmatrix}.$$

We apply the world-space rotation $\Delta\mathbf{R}_t = \mathbf{R}_y(\Delta\phi_t)\,\mathbf{R}_x(\Delta\theta_t)$ about pivot $\mathbf{c}$. The pose of frame $t$ is:

$$\mathbf{R}_t = \Delta\mathbf{R}_t\,\mathbf{R}_0, \mathbf{t}_t = \Delta\mathbf{R}_t\,(\mathbf{t}_0 - \mathbf{c}) + \mathbf{c},$$

$$\mathbf{P}_t = [\mathbf{R}_t \mid \mathbf{t}_t].$$

## 7.5. Down-Weighting of Generated Frames

To reduce the impact of the unreliability over time (see Section 9.4) of generated frames on reconstruction, the loss of generated frames is multiplied by an exponential decay when training visual particles:

$$w_t = w_{\min} + (1 - w_{\min}) \exp(-k\,(t - t_0)),$$

where $t$ is the frame index, $t_0 = 0$, $k = 0.02$, $w_{\min} = 0.0$.

## 7.6. Training Methods

using position-based fluid (PBF) simulation [27, 28] with an incompressibility constraint.

### 7.6.1. Learnable Buoyancy Strength

In well-controlled scenes [13], the relative strength between buoyancy and gravity is fixed, whereas in the wild, buoyancy strength is underdetermined. To account for this, we make the buoyancy coefficient in the PBF simulation (Appendix B in [13]) learnable and optimize it jointly with reconstruction.

### 7.6.2. High-Frequency Information Loss

To sharpen fine details, we add a frequency-domain loss during visual-particle training on original input frames only (generated views are excluded due to unreliable high-frequency content). Given an RGB frame $I$ and ground truth $\hat{I}$, we compute per-channel 2D FFTs $F = \mathcal{F}\{I\}$, $\hat{F} = \mathcal{F}\{\hat{I}\}$, and define amplitude $A = |F|$, $\hat{A} = |\hat{F}|$ and phase $\beta = \angle F$, $\hat{\beta} = \angle\hat{F}$. The loss is computed as the average absolute difference of amplitude and phase over all frequency bins and RGB channels:

$$\mathcal{L}_{\mathrm{freq}} = \mathrm{mean}(|A - \hat{A}|) + \mathrm{mean}(|\beta - \hat{\beta}|).$$

We apply a linear warm-up weight $w_t = \min\left(1, \frac{\mathrm{iter}}{t_{\mathrm{warmup}}}\right)$ and scale by $\lambda_{\mathrm{freq}}$:

$$\mathcal{L}_{\mathrm{FFT}} = \lambda_{\mathrm{freq}}\,w_t\,\mathcal{L}_{\mathrm{freq}},$$

here we set $\lambda_{\mathrm{freq}} = 0.001$.

### 7.6.3. Total Loss

During physical-particle training, we add a learnable buoyancy coefficient in the PBF simulation (Sec. 7.6.1). During visual-particle training, we incorporate the high-frequency loss (Sec. 7.6.2). All remaining losses—including photometric, depth consistency, smoothness, and PBF constraints—are kept identical to FluidNexus [13].

## 7.7. Training and Inference Cost

We report the time and memory costs of our pipeline across all datasets. The corresponding configurations are summarized in Table 6. Even the maximum runtime is significantly lower than that of FluidNexus (which incurs a large training cost due to two generative models) and Hyfluid (which requires about 30 GPU hours to train its three-stage pipeline).

# 8. Alternative Components

**VideoMaMa [25] for Smoke Extraction.** We replace the dehazing model with VideoMaMa and additionally refine masks for thick smoke before testing. This setting changes the evaluation ground truth during testing; therefore, the numbers in Table 5 only indicate results under this alternative setup and are not directly comparable to our main results.

**VGGT [38] for Initialization.** We test VGGT as a replacement for DUSt3R in both point initialization and pose estimation. Table 5 summarizes the experimental results.

Table 5. PSNR comparison (higher is better) of alternative components. NVS = Novel View Synthesis, FP (IV) = Future Prediction (Input View), and FP (NV) = Future Prediction (Novel View).

| | FLAME | Pixabay | | | | Synthetic | |
| | | City | Valley | Forest | NVS | FP (IV) | FP (NV) |
|---|---|---|---|---|---|---|---|
| VideoMaMa [25] | 22.23 | 20.27 | 21.49 | 18.07 | 28.05 | 22.82 | 20.72 |
| VGGT [38] | **27.80** | 24.16 | **26.96** | **19.43** | 29.18 | 24.63 | 24.28 |
| Ours | 22.88 | **24.68** | 20.42 | 17.91 | **29.78** | **25.26** | **25.04** |

**Trajectorycrafter [46] for generation.**   We also test other video-to-video generation models, such as TrajectoryCrafter. However, these models do not explicitly constrain the view center under user-defined camera trajectories, so the camera target often drifts over time. As a result, generated views may gradually shift away from the intended scene center and fail to provide stable multi-view supervision. Representative failure cases are shown in Figure 11. Compared with these alternatives, SV4D 2.0 better matches our requirement of maintaining a stable view center while following custom camera motion.

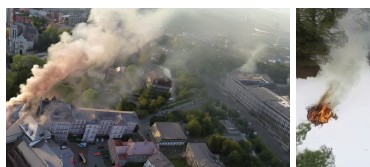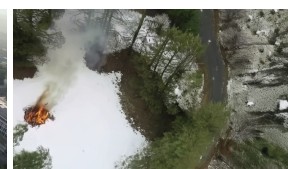

Figure 11. Failure examples of TrajectoryCrafter under custom camera trajectories. The view center drifts, causing the main smoke region to deviate.

**Original FluidNexus [13] and HyFluid [45].**   We evaluate HyFluid [45] and the original FluidNexus [13] as alternative components, but both are less suited to our in-the-wild setting. **FluidNexus** is mainly validated in controlled lab captures where the gravity direction and buoyancy strength are well-defined. In wild-smoke videos, the scene orientation is often unknown and plume dynamics are more diverse. A fixed buoyancy or gravity configuration can therefore produce an incorrect force balance and degrade reconstructions. This motivates our *learnable buoyancy*, which is jointly optimized with the reconstruction. FluidNexus also depends on several scene-dependent hyperparameters, such as domain bounds and validity thresholds. These are easier to specify in a bounded lab volume but hard to define in unconstrained outdoor videos. We address this with a robust *automatic initialization* that provides a reasonable starting volume and scale without manual range tuning. **HyFluid** uses a NeRF-style monocular reconstruction backbone, which is often less stable on in-the-wild videos. This instability reduces the benefit of our local pose perturbation strategy. HyFluid also requires a costly multi-stage training procedure, which leads to substantially longer runtimes.

## 9. More Results

### 9.1. Training and Inference Cost

Table 6 in the App. 7.7 reports the per-stage GPU hours. Across datasets and resolutions, our pipeline reconstructs smoke from in-the-wild videos in 4.47 GPU-hours on average among all 5 videos. Notably, with the DUSt3R-based pose initialization, smoke can be localized without seeding a large number of particles, reducing the Gaussian-particle training stage by 1.5 GPU-hours.

### 9.2. Physical Plausibility

**Divergence Metric.**   To quantify the physical plausibility of the reconstructed motion, we measure the velocity divergence on the predicted Gaussian particles. Given particle positions $\{\mathbf{x}_i\}_{i=1}^{N}$ and velocities $\{\mathbf{v}_i\}_{i=1}^{N}$ at each frame, we estimate the local velocity Jacobian by fitting a first-order model within a fixed physical radius $r$:

$$\mathbf{v}(\mathbf{x}) \approx \mathbf{v}_i + \mathbf{J}_i(\mathbf{x}-\mathbf{x}_i), \qquad \mathbf{J}_i = \left(\mathbf{A}^\top \mathbf{W}\mathbf{A}+\lambda\mathbf{I}\right)^{-1}\mathbf{A}^\top \mathbf{W}\mathbf{B}, \tag{7}$$

where rows of $\mathbf{A}$ and $\mathbf{B}$ are $(\mathbf{x}_j - \mathbf{x}_i)^\top$ and $(\mathbf{v}_j - \mathbf{v}_i)^\top$ for neighbors $j$ satisfying $\|\mathbf{x}_j - \mathbf{x}_i\| \le r$, and $\mathbf{W}$ is a diagonal matrix of Gaussian weights $w_{ij} = \exp(-\|\mathbf{x}_j-\mathbf{x}_i\|_2^2/(2\sigma^2))$ with $\sigma$ tied to $r$. The divergence at particle $i$ is then computed as the trace:

$$(\nabla \cdot \mathbf{v})_i = \text{tr}(\mathbf{J}_i). \tag{8}$$

We report the mean of $|\nabla \cdot \mathbf{v}|$ in Table 7. Our method demonstrates better physical plausibility.

### 9.3. Smoke Extraction

**Extracted Smoke Examples.**   Figure 12 shows representative results of our smoke extraction across several in-the-wild videos. For light smoke, we apply dehazing to remove background leakage; for dense smoke, we use the masked result directly as the clean foreground.

**Results without Smoke Extraction.**   We further train HyFluid [45] and FluidNexus [13] on inputs without our smoke extraction method. The results are shown in Table 8.

### 9.4. Generated Videos from SV4D 2.0

We previously noted that the quality of generated multi-view frames from SV4D 2.0 decays over time. Figure 13 shows early and late frames of the sequence. As time progresses, the smoke structure gradually collapses.

### 9.5. SSIM and LPIPS

Beyond PSNR, we further report the structural similarity index measure (SSIM) in Table 9 and the perceptual metric LPIPS [49] in Table 10. Our method is still shown to perform better based on these two additional metrics.

Table 6. Video resolutions ($W \times H \times T$) and runtime/memory costs of our pipeline. [†]Total time is the sum over all stages; memory reports the peak GPU memory across stages.

| Dataset | Synthetic ($1080 \times 1920 \times 270$) | | FLAME ($1920 \times 1080 \times 270$) | | Pixabay ($1920 \times 1080 \times 270$) | |
|---|---|---|---|---|---|---|
| Stage | Time (GPU-hours) | Mem. (GB) | Time (GPU-hours) | Mem. (GB) | Time (GPU-hours) | Mem. (GB) |
| Smoke Extraction | 0.03 | 5 | 0.03 | 5 | 0.03 | 5 |
| Background Removal | 0.02 | 10 | 2.07 | 10 | 0.02 | 10 |
| Multi-Views | 2.35 | 30 | 2.56 | 30 | 2.55 | 30 |
| Gaussian Particles | 1.05 | 10 | 1.18 | 15 | 1.77 | 20 |
| Total[†] | 3.45 | 30 | 5.84 | 30 | 4.37 | 30 |

Table 7. Physical plausibility measured by velocity divergence on reconstructed particles. Lower is better.

| | FLAME | City | Valley | Forest | Synthetic |
|---|---|---|---|---|---|
| FluidNexus | 1.611 | 1.454 | 0.845 | 0.795 | 1.381 |
| Ours | **0.866** | **0.635** | **0.784** | **0.751** | **0.740** |

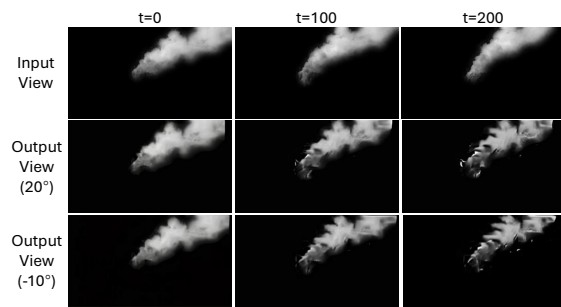

Figure 13. SV4D 2.0 multi-view generation over time. Later frames exhibit structural collapse.

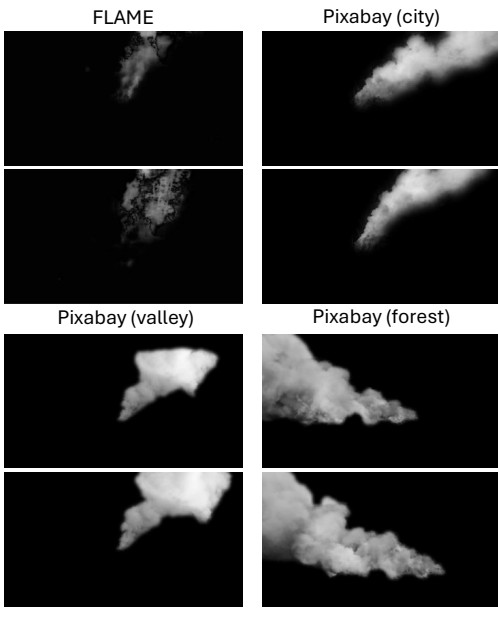

Figure 12. Visualizations of extracted smoke from wild videos.

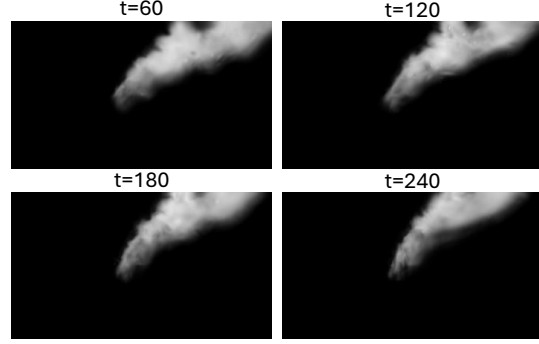

Figure 14. Novel view synthesis on wild videos. Our reconstructions preserve stable smoke structure across views.

Table 8. Comparing PSNR (higher is better) of methods with/without our smoke extraction (SE) method.

| | Future Prediction (Input View) | | | |
|---|---|---|---|---|
| | FLAME | City | Valley | Forest |
| HyFluid | 10.37 | 12.37 | 11.79 | 9.88 |
| HyFluid (w. our SE) | 21.67 | 23.24 | 12.43 | 13.70 |
| FluidNexus | 6.67 | 8.78 | 9.63 | 4.46 |
| FluidNexus (w. our SE) | 21.78 | 24.18 | 16.44 | 14.63 |
| Ours | **22.88** | **24.68** | **20.42** | **17.91** |

Table 9. Comparing SSIM (higher is better) of smoke reconstruction by different methods on videos collected from FLAME [33] and Pixabay.

| | Future Prediction (Input View) | | | |
|---|---|---|---|---|
| | FLAME | City | Valley | Forest |
| HyFluid [45] | 0.87 | 0.55 | 0.79 | 0.46 |
| FluidNexus [13] | 0.88 | 0.84 | 0.76 | 0.70 |
| Ours | **0.92** | **0.89** | **0.86** | **0.80** |

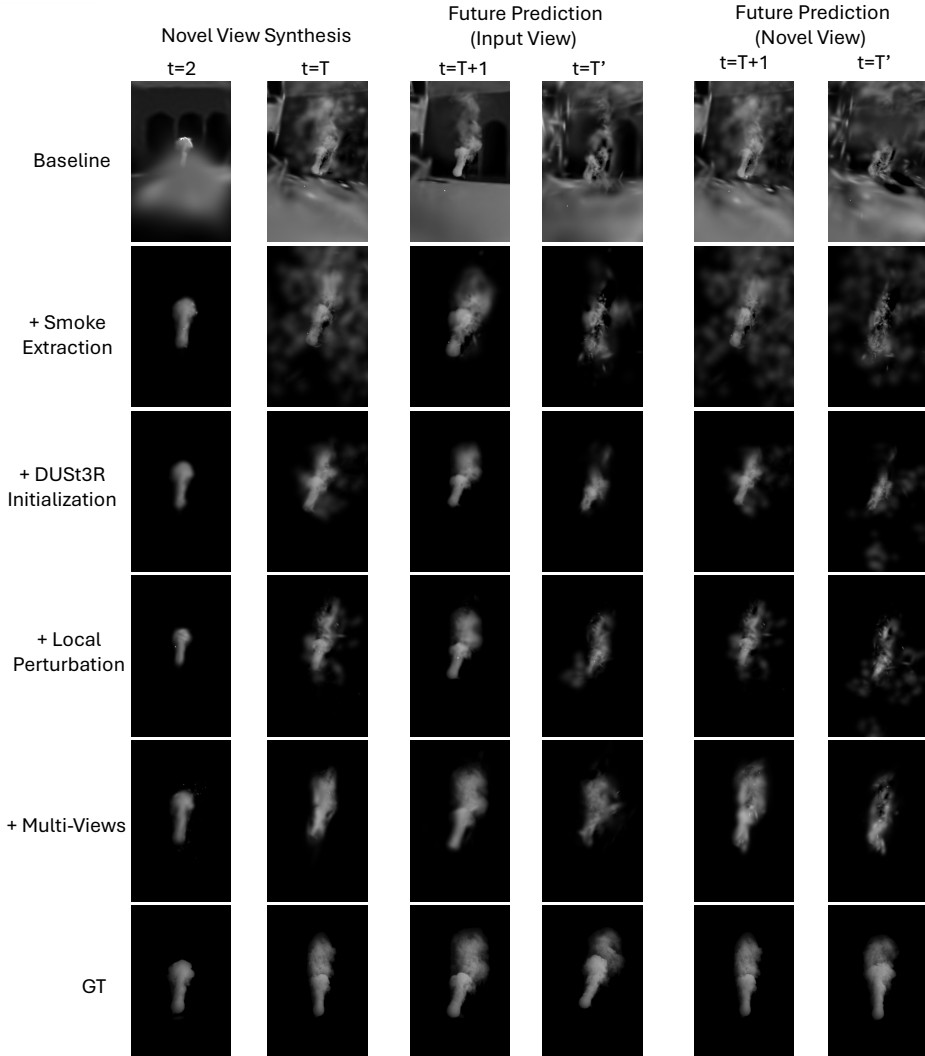

Figure 15. Ablation study of novel view synthesis and future predictions on synthetic smoke videos. *Novel view* uses the camera pose at $t = 1$, and *input view* means the camera poses along the source video. Training uses $T = 240$ frames, and the unseen future extends to $T' = 270$ frames. *GT*: Ground Truth.

Table 10. Comparing LPIPS (smaller is better) of smoke reconstruction by different methods on videos collected from FLAME [33] and Pixabay.

|  | Future Prediction (Input View) | | | |
|  | FLAME | City | Valley | Forest |
|---|---|---|---|---|
| HyFluid [45] | 0.13 | 0.33 | 0.18 | 0.36 |
| FluidNexus [13] | 0.16 | 0.14 | 0.20 | 0.31 |
| Ours | **0.09** | **0.12** | **0.13** | **0.24** |

### 9.6. Visualizations of Novel View Synthesis on Wild Videos

Figure 14 renders novel views for several wild videos. For all frames, the novel-view camera is fixed to the pose of the first frame of the training video. The reconstructed smoke remains stable over time.

### 9.7. Visualizations for Ablation Study on Synthetic Data

We provide additional qualitative ablations (Figure 15) complementing the quantitative table in the main paper.

### 9.8. Failed Examples Using Pretrained Dehazing Model

We also test the extraction performance of the pretrained dehazing model [35]. As shown in Figure 16, the pretrained model fails to remove the light and spatially thin smoke, leaving most of the haze untouched. This prevents us from

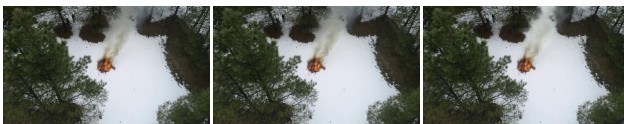

Figure 16. Failure cases of pretrained dehazing model on light outdoor smoke. From left to right: input image, output of the pretrained model (fails to remove light smoke), and output of the finetuned model (significantly clearer background, enabling smoke extraction).

separating the smoke component using simple priors. In contrast, the finetuned dehazing model successfully removes most of the smoke and reveals a clearer background, which subsequently enables reliable extraction of the smoke layer using the dark channel prior.

### 9.9. Human Validation

We conduct a perceptual study to evaluate GIF visualizations of reconstructed smoke dynamics. We collect 14 responses via an online survey. For each question, participants are shown a target GIF and anonymized candidate results (labeled as A/B/C), and are asked to *choose the result closest to the target* or select *Not sure*. We evaluate 7 settings: four in-the-wild sequences (FLAME, Forest, City, Valley) and three synthetic settings (Novel View Synthesis (NVS), Future Prediction in Input View (FP (IV)), and Future Prediction in Novel View (FP (NV))).

We report the percentage of votes for each method in Table 11. As shown in the table, our method is preferred in all evaluated settings.

Table 11. Human preference (%, higher is better) on GIF visual-quality survey (14 respondents). Participants selected the result closest to the target.

| | In-the-wild | | | | Synthetic | | |
| | Forest | City | FLAME | Valley | FP (IV) | NVS | FP (NV) |
|---|---|---|---|---|---|---|---|
| HyFluid | 0.0 | 0.0 | 7.1 | 0.0 | 0.0 | 0.0 | 0.0 |
| FluidNexus | 0.0 | 7.1 | 0.0 | 0.0 | 7.1 | 14.3 | 0.0 |
| Ours | **100.0** | **92.9** | **85.7** | **100.0** | **85.7** | **71.4** | **71.4** |
| Not sure | 0.0 | 0.0 | 7.1 | 0.0 | 7.1 | 14.3 | 28.6 |

## 10. Broader Impact

This paper presents work whose goal is to advance the field of machine learning. There are many potential societal consequences of our work, none of which we feel must be specifically highlighted here. More specifically, our method may support visual effects, simulation, education, and research by making editable dynamic smoke assets easier to obtain from ordinary videos. Potential risks include misuse for creating misleading smoke- or disaster-related visual content, and reconstructed dynamics from a single video should not be treated as ground-truth evidence for safety-critical or forensic decisions.