# OpenReview forum: "WildSmoke: Ready-to-Use Dynamic 3D Smoke Assets from a Single Video in the Wild"
_thecvf.com/CVPR/2026/Workshop/3D4S — CVPR 2026 Workshop 3D4S Poster_

### Official Review · Reviewer_7BBe · 2026-04-19
**A compelling pipeline for dynamic 3D smoke reconstruction from in-the-wild videos**

**Rating:** 7
**Confidence:** 4

**Review:**

## Summary

This paper presents WildSmoke, a pipeline for reconstructing dynamic 3D smoke from a single in-the-wild video. The method addresses several key challenges in unconstrained settings, including background noise, unknown camera poses, and limited viewpoints. To overcome these difficulties, the authors design a complete system that integrates segmentation, dehazing, pose estimation, multi-view generation, and Gaussian-based dynamic reconstruction, followed by simulation refinement.

The goal is to recover temporally coherent 4D smoke representations that can be further used for downstream simulation and editing. The method is evaluated on both synthetic datasets and real-world videos, demonstrating improved reconstruction quality and plausible dynamic behavior.

---

## Strengths

1. **Interesting and challenging problem.**
   Reconstructing dynamic 3D smoke from a single real-world video is a highly challenging and relatively underexplored problem with clear relevance to both vision and graphics communities.

2. **Strong and complete system design.**
   The proposed pipeline is carefully constructed to address multiple practical challenges in real-world scenarios. The integration of segmentation, pose estimation, and multi-view synthesis into a unified framework is well thought out and effective.

3. **Convincing experimental results.**
   The method demonstrates both quantitative improvements (e.g., PSNR gains) and strong qualitative results on synthetic and real-world data, indicating that the approach is effective in practice.

4. **Practical impact and usability.**
   The ability to reconstruct dynamic smoke and support downstream simulation makes the work valuable for applications such as visual effects, content creation, and physical modeling.

5. **Good balance between engineering and problem formulation.**
   While the method builds upon existing components, the way these elements are combined to solve a difficult real-world problem is non-trivial and demonstrates solid technical insight.

---

## Weaknesses

1. **Limited methodological novelty in individual components.**
   Most components in the pipeline are adapted from existing methods, and the novelty mainly lies in their integration.

2. **Dependence on multiple modules.**
   The performance may rely on the robustness of several external components, which could affect generalization in more challenging scenarios.

3. **Limited discussion of failure cases.**
   A more thorough analysis of failure modes would strengthen the paper.

---

## Clarity

The paper is well written and clearly organized. The pipeline is easy to follow, and figures effectively illustrate the method and results.

---

## Overall Assessment

Overall, this paper presents a compelling and well-executed approach to a challenging and relevant problem. Despite relying on existing components, the integration is non-trivial and results in a system that performs well in practice. The work is both technically solid and practically useful, and it provides a valuable contribution to dynamic scene reconstruction from real-world data.

---

## Recommendation

**7: Accept**

---

### Official Review · Reviewer_jdXP · 2026-04-20
**Useful dynamic smoke asset pipeline, but ready-to-use claims need stronger support**

**Rating:** 6
**Confidence:** 4

**Review:**

This paper addresses an interesting and highly relevant problem for 3D4S. It proposes a practical pipeline to address many challenging components and the results/ablation studies also support its effectiveness. Besides, I also like how the authors evaluate on both real and synthetic videos.

The paper is well written and easy to follow. I especially like how each stage of the pipeline is motivated by a specific difficulty in in-the-wild smoke reconstruction. This makes the system feel well thought out, rather than just a collection of separate modules.

For my concerns: The paper does not fully analyze how potential hallucinations or geometric errors in generated multi-view supervision may bias the reconstructed 3D smoke field, especially on real videos where independent multi-view ground truth is unavailable.

In addition, the main paper provides relatively limited quantitative evidence for simulation readiness or physical plausibility. Since “ready-to-use asset” is a central claim, it would be helpful to include at least some physical/simulation-level metrics in the main paper.

---

### Official Review · Reviewer_twzZ · 2026-04-26
**I consider this work to be technically competent, especially as a system paper. The authors try to solve a difficult problem: recovering editable 4D smoke assets from a single real-world video, where the background is noisy, camera poses are unknown, and the smoke itself is highly non-rigid. The pipeline is not simple. It brings together smoke extraction, dehazing, pose initialization, generated multi-view supervision, Gaussian-particle training, and later simulation.**

**Rating:** 6
**Confidence:** 4

**Review:**

## Recommendation: Borderline Accept

### Quality

I consider this work to be technically competent, especially as a system paper. The authors try to solve a difficult problem: recovering editable 4D smoke assets from a single real-world video, where the background is noisy, camera poses are unknown, and the smoke itself is highly non-rigid. The pipeline is not simple. It brings together smoke extraction, dehazing, pose initialization, generated multi-view supervision, Gaussian-particle training, and later simulation.

That said, I would not describe the technical contribution as a major new reconstruction framework. Much of the strength comes from carefully connecting existing components. This is still valuable, but the paper should be more careful in presenting the contribution as a practical pipeline rather than a fundamentally new algorithm.

The reported PSNR improvement is encouraging

The results suggest that the full pipeline works better than prior methods in the tested settings. My concern is that the test set remains small, so the robustness claim should be stated with some restraint.

### Clarity

The paper is generally readable. The authors explain the three main challenges clearly: noisy backgrounds, unknown camera poses, and the coupling between camera viewpoint and time. Figure 2 is helpful because it shows the full pipeline in a compact way.

some important details are pushed into the supplement. From a reviewer’s perspective, this weakens the main paper. The reader needs more information about component choices, training settings, failure cases, and the behavior of the system when one module fails. In particular, the role of generated multi-view supervision deserves a clearer explanation in the main text, since it is one of the more interesting parts of the method.

### Originality

The originality is moderate. I do not see the paper as introducing a new model in the same sense as a new reconstruction architecture or a new physical solver. Its value lies more in the engineering design: taking tools such as SAM/SegGPT, DehazeFormer, DUSt3R, SV4D, Gaussian splatting, and fluid simulation, then adapting them into a single workflow for in-the-wild smoke.

In my view, the most original part is the attempt to use generated multi-view videos to reduce the coupling between time and camera trajectory. This is a reasonable idea, and it fits the problem well. However, generated views can also introduce wrong geometry or temporal artifacts. The paper should discuss this risk more directly.

### Importance

The problem is meaningful. Editable dynamic smoke assets are useful for VFX, simulation, content creation, and dynamic 3D scene modeling. A method that can start from casual real-world footage is clearly attractive.

The “ready-to-use” claim is appealing, but it is also a little strong. The simulation examples with wind and obstacles are visually interesting, yet they are mostly qualitative. I would be more convinced if the authors included a small user study, an editing comparison, or a more concrete evaluation of physical plausibility.

### Strengths

- The paper studies a practical and underexplored problem: reconstructing 4D smoke assets from a single in-the-wild video.
- The pipeline is complete and application-oriented.
- The use of smoke extraction, dehazing, pose initialization, and multi-view generation is sensible.
- The quantitative results show clear gains over HyFluid and FluidNexus in the reported settings.
- The simulation examples make the output more useful than simple novel-view rendering.

### Weaknesses

- The method depends heavily on existing pretrained tools, so the core algorithmic novelty is limited.
- The real-world evaluation set is small, which makes the robustness claim less convincing.
- The generated multi-view supervision may introduce artifacts, but this issue is not analyzed in enough depth.
- The editing results are mostly visual demonstrations rather than rigorous evaluations.
- Some key implementation and validation details are left to the supplement.

### Final Assessment

I would lean toward acceptance if this is submitted to a workshop focused on dynamic 3D assets, scientific 3D/4D modeling, or practical reconstruction systems. The paper is useful and well motivated, even though it is not a major algorithmic breakthrough. My main recommendation is to reduce the strength of some claims, explain the generated-view component more carefully, and add more evidence for robustness and editability.

---

### Decision · Program_Chairs · 2026-04-28

Accept (Poster)